# Corrosion Behavior of CrFeCoNiV_0.5_ and CrFeCoNiV Alloys in 0.5 M and 1 M Sodium Chloride Solutions

**DOI:** 10.3390/ma16144900

**Published:** 2023-07-08

**Authors:** Chun-Huei Tsau, Hui-Ping Hsiao, Tien-Yu Chien

**Affiliations:** Institute of Nanomaterials, Chinese Culture University, Taipei 111, Taiwan; a125b123@yahoo.com.tw (H.-P.H.); school0952067023@gmail.com (T.-Y.C.)

**Keywords:** CrFeCoNiV_x_, corrosion, potentiodynamic polarization test, electrochemical impedance spectroscopy, critical pitting temperature

## Abstract

The effects of the concentration of NaCl solutions on the corrosion resistance of granular CoCrFeNiV_0.5_ and dendritic CrFeCoNiV high-entropy alloys were studied. The polarization behavior of CoCrFeNiV_0.5_ and CrFeCoNiV alloys in deaerated 0.5 M and 1 M sodium chloride solution at different temperatures was measured by a constant galvanostatic/potentiometric. Electrochemical impedance spectroscopy (EIS) was used to examine CoCrFeNiV_0.5_ CrFeCoNiV alloys in 0.5 M and 1 M NaCl solutions. The results indicated that the CoCrFeNiV_0.5_ alloy showed a better corrosion resistance than that of CrFeCoNiV alloy because the dendritic structure of CrFeCoNiV had too many σ/FCC interfaces. The critical pitting temperatures (CPTs) of the alloys under different applied potentials were also analyzed. All of the results proved that CrFeCoNiV_0.5_ alloy had better corrosion resistance in 0.5 M and 1 M NaCl solutions.

## 1. Introduction

The high-entropy-alloy concept has been pointed out to have many benefits, such as cocktail effects, sluggish diffusion, high entropy and severe lattice distortion [1,2]. Therefore, this concept is now widely applied to develop new alloys for specific applications. Researchers select suitable elements and processes to produce new alloys. The strength of AlCoCrFeNiTi alloy was improved because lattice distortion occurred after adding titanium, and the phase transformed from Face-Centered Cubic (FCC) to Body-Centered Cubic (BCC) after adding aluminum [3]. The metallurgy annealing and working treatment of CoCrFeMnNi high-entropy alloy were studied to understand the application potential of this alloy [4]. The mechanical properties of high-entropy alloys were improved through a mechanical alloy process [5], choosing high melting-point elements to develop refractory alloys, such as HfTaTiNbZr, NbMoTaW and VNbMoTaW-based high-entropy alloys [6]. VNbMoTa alloy had excellent room-temperature ductility, its fracture strain was more than 25%, and its compressive yield strength was 811 MPa at 1000 °C [7]. The strength of NbMoTaWVTi high-entropy refractory alloy was enhanced by a mechanical process and spark plasma sintering [8]. This concept is also used to develop new alloys that have a good corrosion resistance. The major elements stainless steel, iron, nickel, chromium and cobalt are frequently chosen to be the major elements to develop new alloys with a good corrosion resistance. Hence, many high-entropy alloys based on the CoCrFeNi-system have been investigated. The addition of chromium significantly enhanced the corrosion resistance in 1 M HNO_3_ and 1 M HCl solutions [9]. The corrosion resistance of Al_x_CrFeCoNi high-entropy alloys was enhanced in 3.5% NaCl solution by increasing the Al and Cr content [10]. The corrosion resistance of (CoCrFeNi)_100−x_Mo_x_ alloys was improved by adding molybdenum in a 3.5% NaCl solution [11]. The corrosion resistance of Q235 alloy steel coated with AlCr*_x_*NiCu_0.5_Mo high-entropy alloy could be significantly improved in 3.5% NaCl solution [12]. Adding Zr into CoCrFeMnNi high-entropy alloys could improve the mechanical properties [13]. The corrosion resistance of 304 stainless steel coated with NbTiAlSiZrN_x_ alloys could also enhance the corrosion resistance in a 0.5 M H_2_SO_4_ solution [14]. CoCrFeNiSn possessed a good passivation in NaCl solution when compared with stainless steels [15]. Corrosion resistance was exhibited by 304 stainless steel in 0.5 M H_2_SO_4_ solution when coating NbTiAlSiZrN_x_ alloy on its surface [14]. Different microstructures influenced the corrosion properties, and the differences of ultrafine-grained and coarse-grained structures on the corrosion behavior of CoCrFeMnNi high-entropy alloys in 1 M H_2_SO_4_ solution and 3.5% NaCl solution were also investigated [16]. The Nb content changed the microstructures of CrFeCoNiNb_x_ alloys, and the corrosion resistance of CrFeCoNiNb_x_ alloys was also influenced. Increasing the Nb content of CrFeCoNiNb_x_ alloys would increase the hardness of the alloys and decrease the corrosion resistance [17].

The microstructures and hardness of as-cast CrFeCoNiV_0.5_ and CrFeCoNiV alloys were studied in our previous study [18], and the corrosion resistance of CrFeCoNiV_0.5_ and CrFeCoNiV alloys in 1 M H_2_SO_4_ and 1 M HCl solutions was included. The SEM micrographs of CrFeCoNiV_0.5_ and CrFeCoNiV alloys are shown in Figure 1 [18]. The CrFeCoNiV_0.5_ alloy showed a granular structure with a few σ-phased particles in the FCC matrix. Moreover, the CrFeCoNiV alloy exhibited a dual-phased dendritic structure (σ phase + FCC phase). The σ phase had a higher Cr and V content. Additionally, the FCC phase had a higher Fe, Co and Ni content. The average hardness value of CrFeCoNiV_0.5_ was 146 HV, and that of CrFeCoNiV alloys was 471 HV. Therefore, the different microstructures of these two alloys influenced their hardness. The present study investigated the corrosion properties of these alloys in 0.5 M and 1 M deaerated sodium chloride solutions.

## 2. Materials and Methods

The pure elements of chromium, iron, cobalt, nickel and vanadium were selected for use in this experiment. The purity of each element was higher than 99.9%. The alloys, CrFeCoNiV_0.5_ and CrFeCoNiV were prepared by a vacuum arc melting furnace (Arc-Melter, Leitai Vacuum Co., Ltd., Taoyuan, Taiwan) under an argon atmosphere. Repeat the melting four to five times to ensure the uniformity of the alloying elements. Table 1 lists the nominal compositions, and the actual compositions analyzed by a scanning electron microscope (SEM, JEOL JSM-6335F) with an energy dispersive spectrometer (SEM/EDS) are also listed in Table 1. The specimens were cut from the as-cast alloys and then cold-mounted by epoxy resin. The exposed area of every specimen was 0.19635 cm^2^ (diameter was 0.5 cm). The potentiodynamic polarization curves, electrochemical impedance spectroscopy (EIS) and simulation software NOVA v2.1.4 were measured in an electrochemical instrument (potentiostat/galvanostat, Autolab PGSTAT302N). The three electrodes in the experiment were the counter electrode, reference electrode and working electrode. The reference electrode was a standard silver chloride (Ag/AgCl, SSE), and SSE has a potential of 0.197 V relative to the standard hydrogen electrode (SHE) [19]. The counter electrode was a platinum wire (Pt). The working electrode was the specimen. The solution was continuously bubbling with nitrogen gas for 900 s to remove oxygen. The scan rate was 0.001 V/s. A critical pitting temperature (CPT) system was assembled by using a precision power supply set (Keithley 2400, Keithley Instruments, Cleveland, OH, USA) and a data acquisition system set (Agilent 34970A, Keysight Technologies, Santa Rosa, CA, USA), together with a heating system. According to the 2010 ASTM-G150-99 specification [20], the heating rate of the solution is 1 °C/min, and the temperature when the current density reaches 0.1 mA/cm^2^ is defined as the critical pitting temperature. Reagent-grade sodium chloride and deionized water were used to prepare the 0.5 M and 1 M NaCl solutions. The concentration of 0.5 M NaCl solution (2.922 weight percent) was closed to seawater. The 1 M NaCl solution (5.844 weight percent) was chosen to compare the effect of an increasing concentration of NaCl solution on these alloys.

## 3. Results and Discussion

Figure 2a,b are the potentiodynamic polarization curves of CrFeCoNiV_0.5_ alloy tested in deaerated 0.5 M and 1 M NaCl solutions at 30 °C and 60 °C, respectively. Table 2 lists the important values of the potentiodynamic polarization curves. The anodic polarization curve of an alloy is the portion of the curve whose potential is more positive than the corrosion potential (*E*_corr_). Increasing the concentration of NaCl solution would increase the corrosion potential of CrFeCoNiV_0.5_ alloy; the corrosion current density (*i*_corr_) of CrFeCoNiV_0.5_ alloy also increased. The potentiodynamic polarization curves of CrFeCoNiV_0.5_ alloy show two significantly anodic peaks at 30 °C, and only one anodic peak was observed in the curves at 60 °C. Table 2 also lists the values of the critical current density (*i*_crit_) and passivation potential (*E*_pp_) of the anodic peak for CrFeCoNiV_0.5_ alloy. When the applied potential is less than 0.8 V_SHE_, the passivation current density of CrFeCoNiV_0.5_ alloy tested at 30 °C almost remains constant. However, the passivation current density of CrFeCoNiV_0.5_ alloy varies with the applied potential. The breakdown potential (*E*_b_) of CrFeCoNiV_0.5_ alloy is about 1.2 V_SHE_ due to the reduction reaction of oxygen [19]. The difference between the CrFeCoNiV_0.5_ alloy tested at 30 and 60 °C is the passivation area. The passivation current density of the CrFeCoNiV_0.5_ alloy tested at 60 °C increased when increasing the applied potential. The passivation region of CrFeCoNiV_0.5_ alloy also had a breakdown potential (*E*_b_) of about 1.2 V_SHE_. Furthermore, the corrosion current density, the critical current density of the anodic peak, increased for these alloys when compared to the values tested at 30 °C, because the electrochemical reaction improved with the test temperature.

Figure 3 shows the SEM micrograph of CoCrFeNiV_0.5_ alloy after a potentiodynamic polarization test in 1 M deaerated NaCl solution at 30 °C. The CoCrFeNiV_0.5_ alloy has a granular structure, and many pitting holes are observed from the morphology after the potentiodynamic polarization test. The pitting holes were located on the grain boundaries and subgrain boundaries. However, the morphology of CoCrFeNiV_0.5_ alloy shows a mild corrosion type after the potentiodynamic polarization test in deaerated 1 M NaCl solution; many pitting holes are shallow.

Figure 4 shows the Nyquist plot, Bode plot and equivalent circuit diagram by electrochemical impedance spectroscopy (EIS) of CoCrFeNiV alloy tested in 1 M NaCl solution, respectively. The starting points of the CoCrFeNiV_0.5_ alloy tested in 0.5 M and 1 M NaCl solutions are very close in the Nyquist diagram, as shown in Figure 4a. The radius of the semicircle of the CoCrFeNiV_0.5_ alloy tested in 0.5 M NaCl solution is greater than that tested in 1 M NaCl solution; this indicates that the polarization resistance (R_p_) of the CoCrFeNiV_0.5_ alloy tested in 0.5 M NaCl solution is greater than that tested in 1 M NaCl solution. The concentration of ions in 1 M NaCl solution was more than that in the 0.5 M NaCl solution, and the polarization resistance for CrFeCoNiV0.5 alloy tested in 1 M NaCl solution was thus lower than that tested in 0.5 M NaCl solution. Therefore, the corrosion current density for CrFeCoNiV_0.5_ alloy tested in 0.5 M NaCl solution was lower than that tested in 1 M NaCl solution. The values of the solution resistance (R_s_) of CoCrFeNiV_0.5_ alloy tested in 0.5 M and 1 M NaCl solutions are close. Table 3 lists the solution resistance (R_s_) and polarization resistance (R_p_) values of CoCrFeNiV_0.5_ alloys. The corresponding equivalent circuit diagram of CoCrFeNiV_0.5_ alloy in deaerated 1 M NaCl solution is interpolated in Figure 4c, and CoCrFeNiV_0.5_ alloy in deaerated 0.5 M NaCl solution had the same equivalent circuit diagram. In the equivalent circuit diagram, Rp, R_s_, CPE and W are the polarization resistance, solution resistance, constant phase element and Warburg impedance, respectively.

Figure 5a,b display the potentiodynamic polarization curves of CrFeCoNiV alloy tested in 0.5 M and 1 M deaerated NaCl solutions at 30 °C and 60 °C, respectively. Table 4 lists the data from the potentiodynamic polarization curves. Increasing the concentration of NaCl solution would slightly decrease the corrosion potential of CrFeCoNiV alloy, and this result is different when compared with CrFeCoNiV_0.5_ alloy. Increasing the testing temperature and the concentration of NaCl solution would result in increasing the corrosion current density (*i*_corr_) of CrFeCoNiV alloy in NaCl solution. The potentiodynamic polarization curves of CrFeCoNiV alloy show anodic peaks at 30 °C. No significantly anodic peak was observed in the curves tested at 60 °C. The values of the critical current density (*i*_crit_) and passivation potential (*E*_pp_) of the anodic peak of CrFeCoNiV alloy are also listed in Table 4. The difference between the CrFeCoNiV alloy tested at 30 and 60 °C is the passivation regions. The passivation current density of CrFeCoNiV alloy increased when increasing the applied potential while the testing temperature was 60 °C. The passivation region of the CrFeCoNiV alloy had a breakdown potential (*E*_b_) of about 1.2 V_SHE_ when it was tested at 30 °C, but the breakdown potential of the passivation region of CrFeCoNiV alloy decreased slightly to about 1 V_SHE_ when the testing temperature increased to 60 °C. Furthermore, the corrosion current density and the critical current density of the anodic peak all increased with an increasing testing temperature.

Figure 6 shows the SEM micrograph of CoCrFeNiV alloy after a potentiodynamic polarization test in deaerated 1 M NaCl solution at 30 °C. The CoCrFeNiV alloy has a dendritic structure, and many pitting holes are observed from the morphology after the test. The pitting holes were located on the σ/FCC interfaces. However, the morphology of CoCrFeNiV alloy after the potentiodynamic polarization test in deaerated 1 M NaCl solution has severe corrosion when compared with that of CoCrFeNiV_0.5_ alloy.

The Nyquist plot and Bode plot by electrochemical impedance spectroscopy (EIS) of CoCrFeNiV alloy tested in 0.5 M and 1 M NaCl solutions are shown in Figure 7a,b, respectively. The radius of the semicircle of the CoCrFeNiV alloy tested in 0.5 M NaCl solution is greater than that of the CoCrFeNiV alloy tested in 1 M NaCl solution; this also means that the polarization resistance (R_p_) of CoCrFeNiV alloy tested in 0.5 M NaCl solution is greater than that of the CoCrFeNiV alloy tested in 1 M NaCl solution. Therefore, the corrosion current density for CrFeCoNiV alloy tested in 0.5 M NaCl solution was lower than that tested in 1 M NaCl solution. Table 5 lists the values of the solution resistance (R_s_) and polarization resistance (R_p_) for CoCrFeNiV alloys. The equivalent circuit diagrams of CoCrFeNiV alloy tested in deaerated 0.5 and 1 M NaCl solutions were the same as for the CoCrFeNiV_0.5_ alloy.

Figure 8 shows the critical-pitting-temperature test results of as-cast CoCrFeNiV_0.5_ and CrFeCoNiV alloys tested in 1 M NaCl solution under different applied potentials. The pitting resistance of CoCrFeNiV_0.5_ alloy was better than that of the CoCrFeNiV alloy. The critical pitting temperature of the CoCrFeNiV_0.5_ and CrFeCoNiV alloys significantly decreased when increasing the applied potential. Figure 9 shows the relationship of the critical pitting temperature of CrFeCoNiV_0.5_ and CrFeCoNiV alloys with the concentration of NaCl solution and applied potential. Increasing the applied potential and the concentration of NaCl solution would decrease the critical pitting temperature of the alloy. The CrFeCoNiV_0.5_ alloy exhibits a good corrosion resistance in NaCl solution because of its granular structure. Increasing the σ/FCC interface in the alloys would decrease the corrosion resistance in NaCl solution. Table 6 lists the critical pitting temperature of the alloys tested in the NaCl solutions under different applied potentials.

The corrosion current density refers to the electrons released from the corroded alloy; the corrosion rate (mm/y) of CrFeCoNiV_0.5_ and CrFeCoNiV alloys could be calculated from this relationship. Assuming that the average density (*ρ*) of the alloy is the sum of the molar fraction of the element i times its density, *ρ* = ∑*X_i_ρ_i_*. Therefore, the rate of corrosion depth (*D*) can be calculated from the following equation:(1)$$\frac{A\cdot D\cdot \rho }{M}\cdot n\cdot F=A\cdot i_{corr}\cdot t$$
where

*A*: corrosion area;

*D*: corrosion depth of one year;

*ρ*: average density of the alloy;

*M*: average atomic mass;

*n*: number of average valence electrons;

*F*: Faraday constant (96,500 C/mol);

*i*_corr_: corrosion current density; and

*t*: corrosion time (3.1536 × 10^7^ s, one year).

According to Equation (1), the corrosion rates of CrFeCoNiV_0.5_ and CrFeCoNiV alloys in 0.5 and 1 M NaCl solutions under 30 and 60 °C are listed in Table 7. The lower corrosion rates showed that both CrFeCoNiV_0.5_ and CrFeCoNiV alloys possessed a good corrosion resistance in 0.5 M and 1 M NaCl solutions. The corrosion resistances of typical ferrous- and nickel-based alloys are differentiated into six ranks by the corrosion rates [21], which are outstanding (<0.0254 mm/y), excellent (0.0254–0.127 mm/y), good (0.127–0.508 mm/y), fair (0.508–1.27 mm/y), poor (1.27–5.08 mm/y) and unacceptable (>5.08 mm/y). Therefore, CrFeCoNiV_0.5_ alloy tested in 0.5 M NaCl solution at 30 °C possessed an outstanding corrosion resistance (<0.0254 mm/y). Increasing the testing temperature or the concentration of NaCl solution resulted in increasing the corrosion rates of these alloys. However, they still had an excellent corrosion resistance (0.0254–0.127 mm/y) under these conditions.

## 4. Conclusions

The present work studied the corrosion properties of CrFeCoNiV_0.5_ and CrFeCoNiV alloys in 0.5 M and 1 M sodium chloride solutions. The corrosion resistances of the alloys were strongly influenced by their microstructures in these NaCl solutions. The granular CrFeCoNiV_0.5_ alloys exhibited a better corrosion resistance when compared with the CrFeCoNiV dendritic alloy. From the results of the potentiodynamic polarization measurements and EIS, the polarization resistance of CrFeCoNiV_0.5_ alloys was higher than that of CrFeCoNiV alloy; the corrosion current density (*i*_corr_) of CrFeCoNiV_0.5_ alloy was thus lower than that of CrFeCoNiV_0.5_ alloy. Increasing the concentration of NaCl solution and applied potential would decrease the critical pitting temperature (CPT) of the alloys. CrFeCoNIV_0.5_ alloy exhibited the highest CPT, 82 °C, in 0.5 M NaCl solution under an applied potential of 700 mV_SHE_.

## Figures and Tables

**Figure 1 materials-16-04900-f001:**
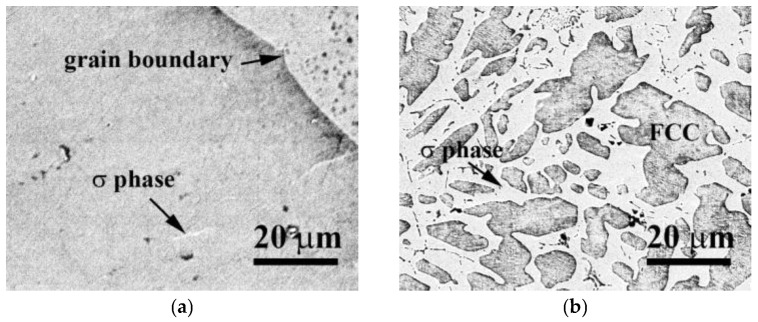
SEM micrographs of as-cast (**a**) CrFeCoNiV_0.5_ alloy and (**b**) CrFeCoNiV alloy [18].

**Figure 2 materials-16-04900-f002:**
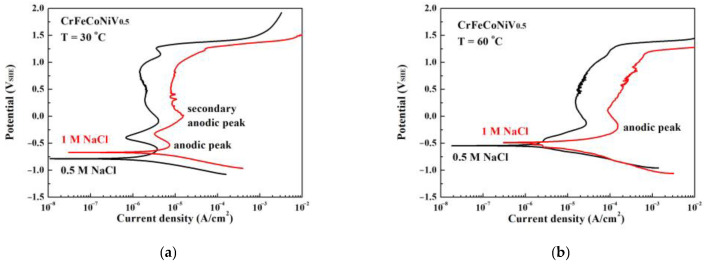
The potentiodynamic polarization curves of as-cast CrFeCoNiV_0.5_ alloy tested in 0.5 M and 1 M deaerated NaCl solutions at (**a**) 30 °C and (**b**) 60 °C.

**Figure 3 materials-16-04900-f003:**
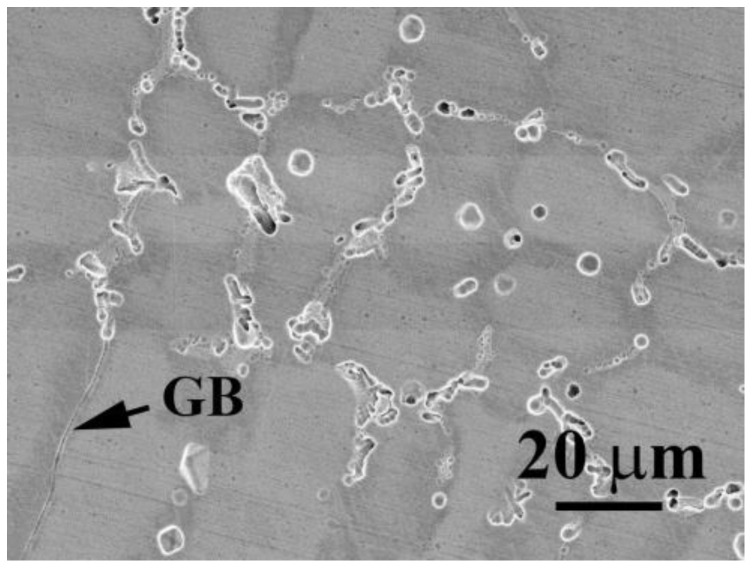
The morphology of CoCrFeNiV_0.5_ alloy after potentiodynamic polarization test in 1 M deaerated NaCl solution at 30 °C.

**Figure 4 materials-16-04900-f004:**
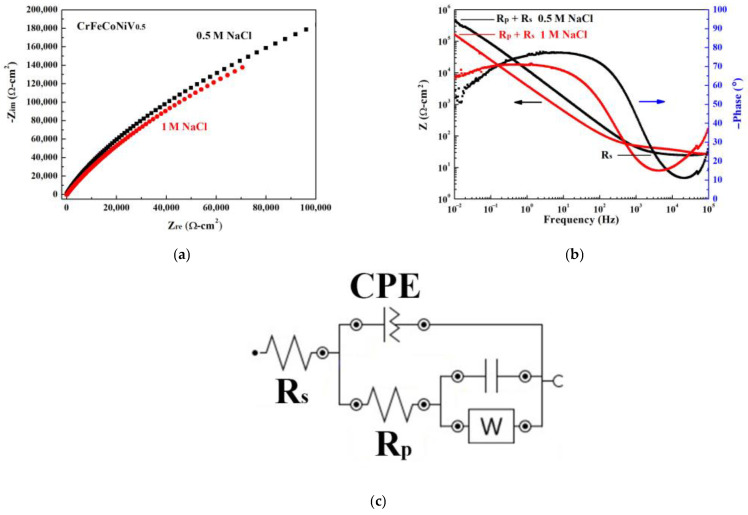
Electrochemical impedance spectroscopy measurements of as-cast CrFeCoNiV_0.5_ alloy. (**a**) Nyquist plot in 0.5 M and 1 M NaCl solutions; (**b**) Bode plot in 0.5 M and 1 M NaCl solutions; and (**c**) corresponding equivalent circuit diagram in 1 M NaCl solution; R_s_ is solution resistance, R_p_ is polarization resistance, CPE is constant phase element, and W is Warburg impedance.

**Figure 5 materials-16-04900-f005:**
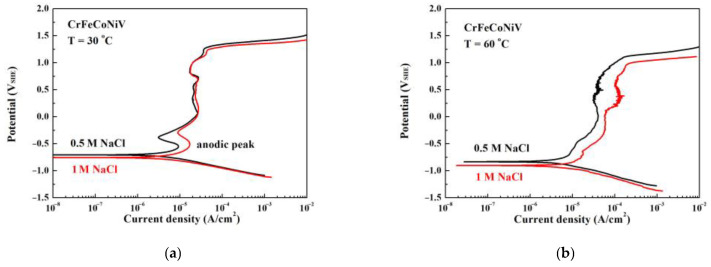
The potentiodynamic polarization curves of as-cast CrFeCoNiV alloy tested in 0.5 M and 1 M deaerated NaCl solutions at (**a**) 30 °C and (**b**) 60 °C.

**Figure 6 materials-16-04900-f006:**
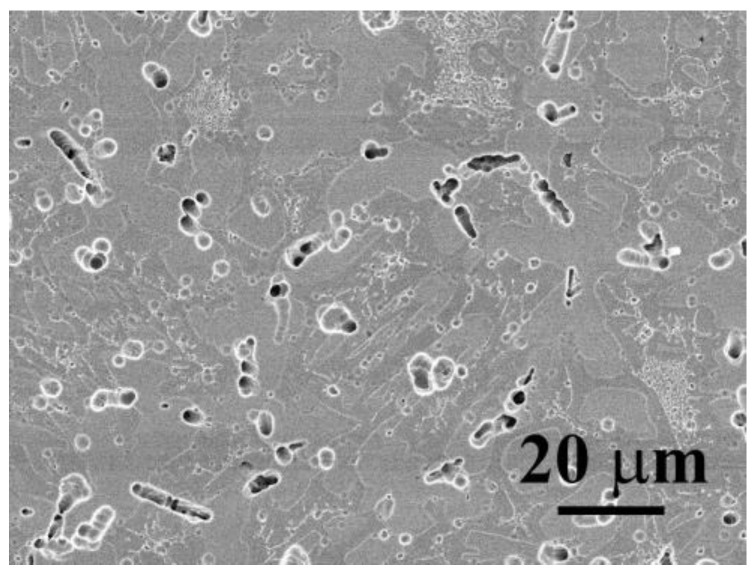
The morphologies of CoCrFeNiV alloy after potentiodynamic polarization tests in 1 M deoxidized NaCl solution at 30 °C.

**Figure 7 materials-16-04900-f007:**
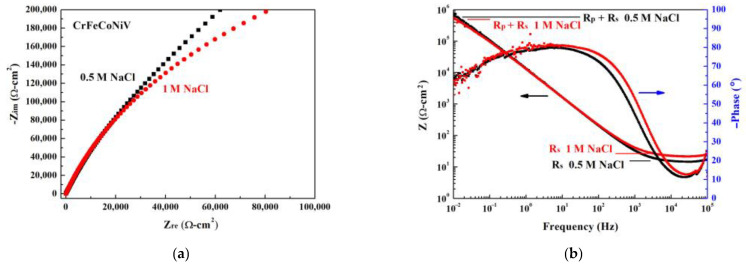
Electrochemical impedance spectroscopy measurements of as-cast CrFeCoNiV alloys in 0.5 M and 1 M NaCl solutions; (**a**) Nyquist plots and (**b**) Bode plots in 1 M and 0.5 M NaCl solutions.

**Figure 8 materials-16-04900-f008:**
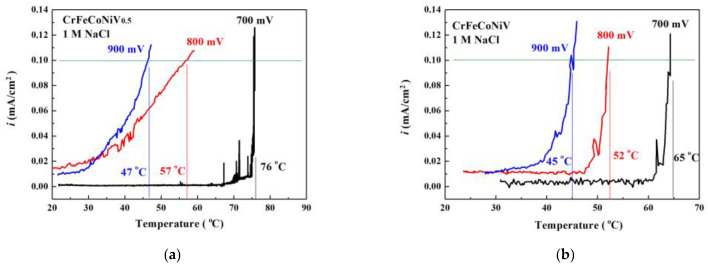
The critical-pitting-temperature test results of as-cast (**a**) CoCrFeNiV_0.5_ and (**b**) CrFeCoNiV alloys tested in 1 M NaCl solution under different applied potentials.

**Figure 9 materials-16-04900-f009:**
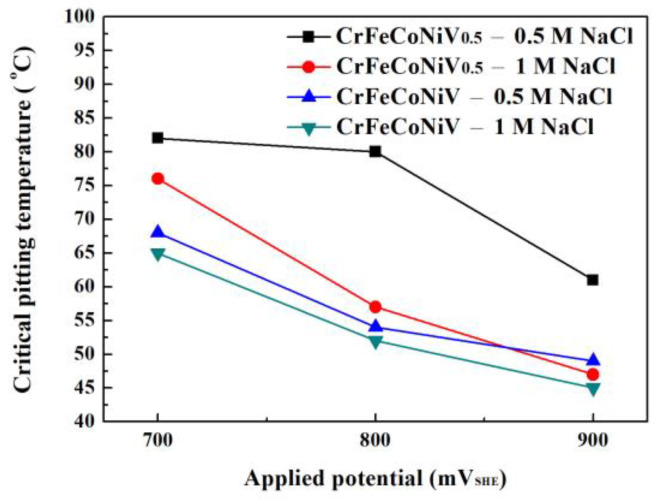
Plot of the relationship of the critical pitting temperatures of CrFeCoNiV_0.5_ and CrFeCoNiV alloys with the concentration of NaCl solutions and applied potentials.

**Table 1 materials-16-04900-t001:** Nominal compositions and actual compositions examined by SEM/EDS of CrFeCoNiV_0.5_ and CrFeCoNiV alloys.

Alloys (at.%)	Cr	Fe	Co	Ni	V
	(wt.%)				
CrFeCoNiV_0.5_					
nominal	20.72	22.25	23.48	23.39	10.16
actual	21.6	22.6	22.3	22.3	11.2
CrFeCoNiV					
Nominal	18.81	20.20	21.32	21.24	18.43
actual	19.4	20.2	20.5	20.2	19.7

**Table 2 materials-16-04900-t002:** Potentiodynamic polarization data of the as-cast CrFeCoNiV_0.5_ alloy in 0.5 M and 1 M deaerated NaCl solution at different temperatures.

	0.5 M NaCl		1 M NaCl	
	30 °C	60 °C	30 °C	60 °C
*E*_corr_ (V_SHE_)	−0.79	−0.67	−0.55	−0.49
*i*_corr_ (μA/cm^2^)	1.80	4.20	2.90	5.40
*E*_pp_ (V_SHE_)	−0.60	−0.53	−0.12	−0.17
*i*_crit_ (μA/cm^2^)	3.80	7.40	2.76	154
*E*_pp2_ (V_SHE_) *	−0.09	0.01	N/A	N/A
*i*_crit2_ (μA/cm^2^) *	4.10	15.3	N/A	N/A

* Secondary anodic peak. N/A means no this data.

**Table 3 materials-16-04900-t003:** The resistances of as-cast CoCrFeNiV_0.5_ alloys in 0.5 M and 1 M deaerated NaCl solutions.

	0.5 M NaCl	1 M NaCl
R_s_ (Ω)	26	27
R_p_ (kΩ)	517	164

**Table 4 materials-16-04900-t004:** Potentiodynamic polarization data of the as-cast CrFeCoNiV alloy in 0.5 M and 1 M deaerated NaCl solution at different temperatures.

	0.5 M NaCl		1 M NaCl	
	30 °C	60 °C	30 °C	60 °C
*E*_corr_ (V_SHE_)	−0.71	−0.83	−0.76	−0.90
*i*_corr_ (μA/cm^2^)	3.50	4.00	5.00	6.00
*E*_pp_ (V_SHE_)	−0.55	N/A	−0.51	N/A
*i*_crit_ (μA/cm^2^)	9.30	N/A	17.1	N/A

**Table 5 materials-16-04900-t005:** The resistances of as-cast CoCrFeNiV alloy in 0.5 M and 1 M deaerated NaCl solution.

	0.5 M NaCl	1 M NaCl
R_s_ (Ω)	14	20
R_p_ (kΩ)	591	470

**Table 6 materials-16-04900-t006:** Critical pitting temperature (°C) of the alloy in 0.5 M and 1 M deaerated NaCl solutions under different applied potentials.

Applied Potential	CrFeCoNiV_0.5_		CrFeCoNiV	
(mV_SHE_)	0.5 M NaCl	1 M NaCl	0.5 M NaCl	1 M NaCl
700	82	76	68	65
800	80	57	54	52
900	61	47	49	45

**Table 7 materials-16-04900-t007:** Corrosion rates of the as-cast CrFeCoNiV_0.5_ and CrFeCoNiV alloys in 0.5 and 1 M NaCl solutions under 30 and 60 °C.

Alloys	0.5 M NaCl (mm/y)		1 M NaCl (mm/y)	
	30 °C	60 °C	30 °C	60 °C
CrFeCoNiV_0.5_	0.016	0.026	0.038	0.048
CrFeCoNiV	0.029	0.033	0.041	0.050

## Data Availability

Not applicable.

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
