# Peer review of "Corrosion Behavior of CrFeCoNiV_0.5_ and CrFeCoNiV Alloys in 0.5 M and 1 M Sodium Chloride Solutions"

_materials, 2023, doi:10.3390/ma16144900_

Round 1

Reviewer 1 Report

I received your article entitled “Corrosion behavior of CrFeCoNiV0.5 and CrFeCoNiV alloys in 0.5M and 1M sodium chloride solutions” for evaluation. I have read your work with attention and interest, but unfortunately I cannot recommend its publication. The reason is that your article is very similar to another article that you yourselves published in Materials magazine (Materials 2022, 15(10), 3639; https://doi.org/10.3390/ma15103639), entitled “ Corrosion Behavior of CrFeCoNiVx (x = 0.5 and 1) High-Entropy Alloys in 1M Sulfuric Acid and 1M Hydrochloric Acid Solutions”. For me there is no new or significant contribution in the subordinate article in relation to the previous article.

Consider that the results presented by you are already well exposed in your previous article and that there is no need to publish a new article with the same information. This can lead to redundancy and confusion in the scientific literature on the subject.

I suggest that you seek to develop new research or approaches on the subject, which can bring new knowledge and advances to the area. Thank you for your understanding and I hope you can submit a new work in the future.

Author Response

Reply: Our previous study focused on the microstructures of the alloys and the corrosion properties in the 1M H2SO4 and 1M HCL solutions. This manuscript discussed the corrosion behavior of the alloys in 0.5M and 1M NaCl solutions. The results of these two manuscripts are different and unpublished. Thank you for your commands.

Reviewer 2 Report

The manuscript is devoted to the study of the effects of the concentration of NaCl solutions on the corrosion resistance of granular  CoCrFeNiV0.5 and dendritic CrFeCoNiV high-entropy alloys. The manuscript is of great theoretical and practical interest, however there are some remarks.

            1. Lines 155-156. Authors write: "The corresponding equivalent circuit diagram of CoCrFeNiV0.5 alloy in deaerated 1M NaCl solution is interpolated in Figure 4c". However, this is inconsistent with the caption under Fig. 4: "Figure 4. Electrochemical impedance spectroscopy measurements of as-cast CrFeCoNiV0.5 alloy in 0.5M and 1M NaCl solutions, "

            2. Figures 4 and 7 show the same equivalent circuit. Why is it necessary to give the same equivalent circuit twice?

            3. It is not clear from Fig. 8 why the critical pitting temperature at different potentials was determined at the same current density of 0.10 ma/cm2. And why, for example, not at 0.04 ma/cm2? Authors should provide explanations.

            4. Line 38. Authors must spell out the abbreviations FCC and BCC.

            5.  Line 32: "...the high melting-point elements to develop refractory alloys to develop the refractory  alloys ..."

            6. Lines 208-209: "The radius of the semicircle of the CoCrFeNiV alloy  tested in 0.5M NaCl solution is greater than that of the CoCrFeNiV0.5  alloy tested in 1M  NaCl solution; " Instead of CoCrFeNiV0.5 should be CoCrFeNiV.

            7. Lines 284-285: "Increasing the concentration of NaCl solution of applied potential would decrease the critical pitting temperature " Refine the phrase.

            8. Lines 278-279: "The corrosion resistance of the alloys strongly were strongly influenced by their microstructures in these NaCl solutions. " Correct the phrase!

Author Response

  1. Lines 155-156. Authors write: "The corresponding equivalent circuit diagram of CoCrFeNiValloy in deaerated 1M NaClsolution is interpolated in Figure 4c". However, this is inconsistent with the caption under Fig. 4: "Figure 4. Electrochemical impedance spectroscopy measurements of as-cast CrFeCoNiV0.5 alloy in 0.5M and 1M NaCl solutions, "

Reply: We modified the statements (line 157-158) and figure caption (line 164-167).

  1. Figures 4 and 7 show the same equivalent circuit. Why is it necessary to give the same equivalent circuit twice?

Reply: We deleted Figure 7c.

  1. It is not clear from Fig. 8 why the critical pitting temperature at different potentials was determined at the same current density of 0.10 ma/cm2. And why, for example, not at 0.04 ma/cm2? Authors should provide explanations.

Reply: The critical pitting temperature of an alloy is defined by ASTM G150-99 (Ref.21), and the current density is 0.10 mA/cm2.

  1. Line 38. Authors must spell out the abbreviations FCC and BCC.

Reply: We added them.

  1. Line 32: "...the high melting-point elements to develop refractory alloys to develop the refractoryalloys ..."

Reply: It was corrected.

  1. Lines 208-209: "The radius of the semicircle of the CoCrFeNiV alloy  tested in 0.5M NaCl solution is greater than that of the CoCrFeNiV5  alloy tested in 1M  NaCl solution; " Instead of CoCrFeNiV0.5should be CoCrFeNiV.

Reply: It was corrected.

  1. Lines 284-285: "Increasing the concentration of NaCl solution of applied potential would decrease the critical pitting temperature " Refine the phrase.

Reply: it was corrected.

  1. Lines 278-279: "The corrosion resistance of the alloys strongly were stronglyinfluenced by their microstructures in these NaCl solutions. " Correct the phrase!

Reply: it was corrected.

Thank you for your commands.

Reviewer 3 Report

Comments article “Corrosion behavior of CrFeCoNiV0.5 and CrFeCoNiV alloys in 0.5M and 1M sodium chloride solutions”

The paper compares the CrFeCoNiV0.5 and CrFeCoNiV HEA in 0.5M and 1M NaCl solutions, indicating a relation between the material microstructure and the corrosion behavior. The granular microstructure obtained for CrFeCoNiV0.5 resulted in a lower mass loss by corrosion than the dendritic microstructure of CrFeCoNiV.

This work needs a few reviews before its acceptance for publishing in Materials, as follows.

Some specific points that should help the authors:

Line 51: It is “1M H2SO4” instead to “1N H2SO4”.

Line 76: Add the furnace manufacturer.

Line 87: Add a space between “0.197” and “V”.

Line 90: Insert a space between “0.001” and “V/s”.

Line 95: Maintain “0.1” and “mA/cm2” in the same line.

Line 124: Insert a space between “30” and “°C”.

Line 124: Insert a space between “60” and “°C”.

Line 171: Insert a space between “30” and “°C”.

Line 171: Insert a space between “60” and “°C”.

Line 190: Insert a space between “30” and “°C”.

Line 190: Insert a space between “60” and “°C”.

Author Response

Some specific points that should help the authors:

Line 51: It is “1M H2SO4” instead to “1N H2SO4”.

Reply: It was 0.5M, and it was modified.

Line 76: Add the furnace manufacturer.

Reply: We added it.

Line 87: Add a space between “0.197” and “V”.

Reply: It’s modified.

Line 90: Insert a space between “0.001” and “V/s”.

Reply: It’s modified.

Line 95: Maintain “0.1” and “mA/cm2” in the same line.

Reply: It’s modified.

Line 124: Insert a space between “30” and “°C”.

Reply: It’s modified.

Line 124: Insert a space between “60” and “°C”.

Reply: It’s modified.

Line 171: Insert a space between “30” and “°C”.

Reply: It’s modified.

Line 171: Insert a space between “60” and “°C”.

Reply: It’s modified.

Line 190: Insert a space between “30” and “°C”.

Reply: It’s modified.

Line 190: Insert a space between “60” and “°C”.

Reply: It’s modified.

Thank you very much.

Round 2

Reviewer 1 Report

The author made changes to the text, taking into account the comments of other reviewers and improving the quality of the work. After reading the revised version of the article, I decided to change my recommendation from rejecting it to accepting it after a small review. Thank you to the author for the response and for clarifying the purpose of this manuscript. However, I believe that the replacement of acidic solutions by saline solutions could have been more justified and accommodated in the text, as it is a relevant factor for the study of alloy resistance. I recommend that the author use bibliometric analysis (allows revealing the evolutionary nuances of a specific field, while shedding light on emerging areas in this field), as it is a good option, allowing to clarify the scientific contribution and citation potential of manuscripts used as reference, in order to make it clearer what were the criteria for choosing NaCl concentrations and how they compare with real conditions of alloy application.

Author Response

The author made changes to the text, taking into account the comments of other reviewers and improving the quality of the work. After reading the revised version of the article, I decided to change my recommendation from rejecting it to accepting it after a small review. Thank you to the author for the response and for clarifying the purpose of this manuscript. However, I believe that the replacement of acidic solutions by saline solutions could have been more justified and accommodated in the text, as it is a relevant factor for the study of alloy resistance. I recommend that the author use bibliometric analysis (allows revealing the evolutionary nuances of a specific field, while shedding light on emerging areas in this field), as it is a good option, allowing to clarify the scientific contribution and citation potential of manuscripts used as reference, in order to make it clearer what were the criteria for choosing NaCl concentrations and how they compare with real conditions of alloy application.

Reply: The reason that we chose the 0.5M and 1M NaCl solutions was added in this manuscript (line 97-100). And we added some statements to describe the superiority of these alloys in NaCl solutions (line 270-278).
